# A Study of Downlink Power-Domain Non-Orthogonal Multiple Access Performance in Tactile Internet Employing Sensors and Actuators

**DOI:** 10.3390/s24227220

**Published:** 2024-11-12

**Authors:** Vaibhav Fanibhare, Nurul I. Sarkar, Adnan Al-Anbuky

**Affiliations:** 1Department of Computer Science and Software Engineering, Auckland University of Technology, Auckland 1010, New Zealand; vaibhav.fanibhare@aut.ac.nz; 2Department of Electrical and Electronic Engineering, Auckland University of Technology, Auckland 1010, New Zealand; adnan.anbuky@aut.ac.nz

**Keywords:** 5G, B5G, power domain, SISO, MIMO, NOMA, tactile Internet

## Abstract

The Tactile Internet (TI) characterises the transformative paradigm that aims to support real-time control and haptic communication between humans and machines, heavily relying on a dense network of sensors and actuators. Non-Orthogonal Multiple Access (NOMA) is a promising enabler of TI that enhances interactions between sensors and actuators, which are collectively considered as users, and thus supports multiple users simultaneously in sharing the same Resource Block (RB), consequently offering remarkable improvements in spectral efficiency and latency. This article proposes a novel downlink power domain Single-Input Single-Output (SISO) NOMA communication scenario for TI by considering multiple users and a base station. The Signal-to-Interference Noise Ratio (SINR), sum rate and fair Power Allocation (PA) coefficients are mathematically derived in the SISO-NOMA system model. The simulations are performed with two-user and three-user scenarios to evaluate the system performance in terms of Bit Error Rate (BER), sum rate and latency between SISO-NOMA and traditional Orthogonal Multiple Access (OMA) schemes. Moreover, outage probability is analysed with varying fixed Power Allocation (PA) coefficients in the SISO-NOMA scheme. In addition, we present the outage probability, sum rate and latency analyses for fixed and derived fair PA coefficients, thus promoting dynamic PA and user fairness by efficiently utilising the available spectrum. Finally, the performance of 4 × 4 Multiple-Input Multiple-Output (MIMO) NOMA incorporating zero forcing-based beamforming and a round-robin scheduling process is compared and analysed with SISO-NOMA in terms of achievable sum rate and latency.

## 1. Introduction

The Tactile Internet (TI) in the Fifth-Generation (5G) communication infrastructure, and Beyond 5G (B5G) has been considered a potential enabler for service categories such as Enhanced Mobile Broadband (eMBB) communication, Massive Machine-Type Communication (mMTC) and Ultra-Reliable Low Latency Communication (URLLC) [1] to allow the interactions between humans, machines and Internet of Things (IoT) devices with audio and/or visual haptic feedback over a certain remote distance [2,3]. The TI communication infrastructure demands significantly high performance concerning ultra-low latency (≤ 1 ms), high bandwidth (30–300) GHz, ultra-high availability (99.999%), ultra-reliability and high data rate support for Augmented Reality/Virtual Reality (AR/VR) applications.

Considering key enabling TI technologies such as Software-Defined Networking (SDN), Network Slicing (NS), Network Function Virtualisation (NFV), Network Coding (NC) and Multiple Access (MA) techniques, the TI communication infrastructure aims to improve their immersive, responsive and interactive services/applications. Thus, these technologies facilitate necessary dynamic infrastructure, optimising resource allocation and utilisation, offloading computational capabilities, and improving spectral efficiency, Quality of Service (QoS) and Quality of Experience (QoE).

According to the Ericsson Mobility Report [4], a significant 28% surge was observed in mobile network data traffic between the 4th quarter of 2022 and 2023, reaching a monthly consumption of 152 exabytes, contributing to approximately 8.5 billion mobile subscribers. In addition, the Cisco annual Internet report (2018–2023) [5] stated that, by 2023, global Internet users were estimated to touch 5.3 billion, constituting 66% of the world’s population. IoT devices will exceed three times the population, totalling 29.3 billion, whereas Machine-to-Machine (M2M) (a.k.a IoT) connections constitute half. The user devices surpassed the total connections, representing 74%, whereas the IoT applications were forecast to have substantial growth, with a 30% Compound Annual Growth Rate (CAGR).

Despite the technological advancements in 5G communication infrastructure compared to First to Fourth Generation (1G–4G) ones, the system capacity and spectral efficiency improvement fall short of the 1000× increase mentioned in the International Mobile Telecommunications—2020 (IMT-2020) vision [6].

To improve the system capacity and spectral efficiency, the mobile and wireless communication infrastructure has employed several MA techniques from 1G to 4G, such as 1G—Frequency Division Multiple Access (FDMA), Second Generation (2G)—Time Division Multiple Access (TDMA), Third Generation (3G)—Code Division Multiple Access (CDMA), and 4G—Orthogonal Frequency Division Multiple Access (OFDMA), or combinations of these MA techniques. These MA techniques are adopted to share the available bandwidth among multiple users and reply on the resource (time/frequency) orthogonalisation, thus assigning different resources to individual users. Moreover, the bandwidth is wasted due to users’ utilisation of separate orthogonal Resource Blocks (RBs) using traditional MA techniques. These MA techniques are examples of Orthogonal Multiple Access (OMA) schemes. Hence, these traditional MA techniques lack the competencies to serve multiple users having sensors and actuators simultaneously, rendering them unsuitable for a 5G communication infrastructure. In general, MA techniques can be structurally classified into OMA for 1G to 4G and Non-Orthogonal Multiple Access (NOMA) for 5G.

As mentioned, one of the high-performance enablers concerning parameters for TI communication infrastructure in 5G, the MA techniques, viz., NOMA, are complementing the 5G New Radio (NR) robust, flexible and scalable framework. However, integrating NOMA can further resolve challenges and serve the demands of the proliferation of users, whether sensors or actuators, by utilising the same RB, according to 3GPP standardisation [7]. This integration facilitates high system capacity, enhances spectral efficiency, supports massive connectivity and reduces latency in the network [8]. This technique concurrently assigns non-orthogonal resources amongst the multiple users whilst having some interference on the receiver side. In contrast, the traditional OMA techniques assign orthogonal resources solely to individual users, thus incurring challenges in achieving the required system capacity and spectral efficiency in 5G.

Figure 1 depicts the pictorial representation of NOMA when compared to OMA. Considering each user a sensor or an actuator, the power with respect to the resource graph is plotted in order to understand how the resources are allocated to the users in both cases. Figure 1a represents the OMA scheme—how the users are orthogonal to each other and have a dedicated/assigned RB for each user ($U_{1}$, $U_{2}$, *…*, $U_{N}$). On the other hand, Figure 1b represents the NOMA scheme—how multiple users share a single RB, as the available bandwidth is not split across the users and assigns a single frequency channel to multiple users. Hence, the bandwidth wastage is minimised and the system throughput is increased since the entire frequency spectrum is accessible to each user for transmission.

Consequently, it can be inferred that the NOMA environment is crucial and allows a dense deployment of sensors and actuators from users to share the assigned single RB, thus increasing the spectral efficiency and capacity compared to traditional OMA. Moreover, NOMA offers improved user fairness and QoS by having flexibility in Power Allocation (PA) and allowing users to be prioritised for stricter latency requirements by allocating them more power for faster data transmission. There are fewer waiting times to access the resources as users are sharing the resources concurrently, so this leads to overall reduced latency in data transmission. Finally, NOMA can effectively support heterogeneous networks with diverse user requirements and traffic characteristics. These benefits suit TI applications [1], such as enhanced mobile broadband, massive connectivity and latency-sensitive applications.

In addition, NOMA techniques can be categorised into Power-Domain NOMA (PD-NOMA) and Code-Domain NOMA (CD-NOMA). In PD-NOMA, multiple users share the same RB. However, each user (whether a sensor or an actuator) is identified or allocated with varying power levels (coefficients) for transmission according to the channel conditions and distance from the Base Station (BS). Users with high channel gain (near BS) are allocated low power levels, whereas users with low channel gain (far from BS) are allocated high power levels. It utilises superposition coding (multiplexing) at the BS and performs Successive Interference Cancellation (SIC) at the receiver to detect and extract the users’ original data signals, and vice versa. In CD-NOMA, multiple users share the same RB and are assigned distinct non-orthogonal codes, such as Low-Density Spreading (LDS), Sparse Code Multiple Access (SCMA) and Multi-User Shared Access (MUSA) [9]. Increasing the number of users in CD-NOMA requires sophisticated algorithms for multi-user reception when non-orthogonal codes are adopted.

Furthermore, in PD-NOMA, both Uplink (UL) and Downlink (DL) transmissions play crucial roles in facilitating seamless message exchange between the user (whether a sensor or an actuator) and the BS. Considering UL PD-NOMA, the simultaneous transmission happens from user to BS using the same RB and is distinguished by varying power levels. Conversely, for DL PD-NOMA, the simultaneous transmission happens from BS to user, where BS assigns the proportion of power levels to multiple users based on the channel conditions and distance between BS and user. Therefore, these PD-NOMA concepts enhance system capacity and optimise spectral efficiency, thus accommodating more sensors and actuators and reducing congestion in the TI communication infrastructure. Based on a sensor or actuator’s priority, channel conditions and power requirements, it can empower low-power communication to conserve energy and extend the battery life with a less frequent need to replace it.

### 1.1. Research Challenges

While the PD-NOMA is broadly considered a promising multiple access technique which enhances the existing 5G-NR framework and B5G, several challenges [10,11] demand further investigation into implementing NOMA to improve capacity, spectral efficiency and latency. Considering resource allocation in NOMA, multiple users have sensors and actuators share the same RB. They are paired based on the distinct (strong and weak) channel gains, resulting in different PAs. At the BS, the user signals are superimposed and transmitted, whereas, at the user’s location, SIC is applied to received user signals to decode and extract the original signal. However, decoding the superimposed signal using SIC considering perfect Channel State Information (CSI) can be challenging. A perfect CSI can pose real-time channel estimation errors, which may degrade the system’s performance [12]. Therefore, new hybrid techniques and algorithms can be developed to yield a practical channel estimation.

In addition, NOMA’s system performance reduction may arise due to an imperfect SIC receiver [13]. This imperfection depends on building efficient hardware to reduce computational complexity. The efficient PA among multiple users can pose another challenge in the NOMA system. Inefficient PA may lead to interference in the received user signal, unfairness among paired users and higher outage probability, thus resulting in performance degradation. Moreover, most researchers focus on adopting a two-user pairing or sensor clustering for easy superimposition at the BS and to reduce SIC complexity at the receiver’s end. To cater to the growing demands of mMTC devices, multi-user pairing strategies or sensor clustering must be formed to take complete advantage of NOMA systems. Finally, due to the heterogeneous wireless network, a need for hybrid-NOMA (NOMA with other multiple access techniques) arises to address the diverse needs of devices through the BS. Integrating NOMA with other multiple access techniques, optimising the system’s performance, can be challenging.

### 1.2. Motivation

PD-NOMA offers several significant benefits over traditional MA techniques, such as enhancing capacity, increasing spectral efficiency, improving user fairness and QoS, mitigating interference and serving multiple users with the same RB, thus addressing the shortcomings of current communication infrastructure and directing future networks. Therefore, PD-NOMA flexible PA enables and prioritises eMBB, mMTC and URLLC TI applications/services with the required resources for seamless operation. However, research in PD-NOMA is needed to improve system performance further and address the aforementioned challenges, such as PA algorithms, hardware impairments, receiver design with imperfect SIC and CSI and the integration of PD-NOMA with other MA techniques to cater to the diverse needs of user devices. Subsequently, it is driven by the need for an efficient, flexible and user-centric communication infrastructure that meets the ever-growing demands of future networks.

### 1.3. Research Contributions

In accordance with the findings from the literature review and to the best of our knowledge, research is still lacking in the field of PD-NOMA for TI applications. The performance analysis and evaluation of PD-NOMA in TI have not been comprehensively conducted in the past. Therefore, in this research, we have developed a novel downlink PD Single-Input Single-Output (SISO) NOMA communication scenario for TI, employing multiple sensors and actuators (e.g. users), focusing on transmitted signals from a BS and processed received signals at the user’s end. Furthermore, the Key Performance Indicators (KPIs) are derived, discussed and evaluated to compare the performance characterisation of the NOMA and OMA schemes. The main contributions of this paper are highlighted below.

We developed an analytical system model incorporating Signal-to-Interference and Noise Ratio (SINR), sum rate, fair Power Allocation (PA) coefficients and latency among the available SISO-NOMA users.We compared and analysed BER for SISO-NOMA and OMA schemes with varying path loss exponent and fixed PA coefficients using two- and three-user scenarios. To this end, we compared the achievable sum rate and latency trends for the SISO-NOMA and OMA schemes for fixed PA coefficients.The outage probability, achievable sum rate and latency trends are compared and analysed for fixed and fair PA coefficients. To this end, the performance trend of outage probability for SISO-NOMA users in varying fixed PA coefficients is also analysed.Finally, the achievable sum rate and latency are compared and analysed between SISO-NOMA and 4 × 4 Multiple-Input Multiple-Output (MIMO) NOMA, incorporating a zero forcing-based beamforming and a round-robin scheduling process.

### 1.4. Structure of the Article

The rest of the article is organised as follows. Section 2 presents the literature review relevant to the NOMA scheme used in the network. Section 3 describes the proposed system model for the downlink PD SISO-NOMA communication scenario for TI. This section also includes the derivation and analysis of SINR, achievable sum rate, fair PA and latency. The performance evaluation is presented in Section 4. The simulation results are also presented in this section. Section 5 concludes our work with future directions. Finally, Table A1 lists the abbreviations and their explanation (see Appendix A).

## 2. Related Work

Several research studies have been conducted on NOMA to enhance system capacity, improve spectral efficiency and reduce latency. In [14], NOMA is adopted on 5G communication infrastructure (test-bed) with the objective of minimising latency and optimising resource allocation for Autonomous Vehicles (AVs). Also, some challenges for implementing and adopting NOMA for 5G-based applications are highlighted for researchers and AV manufacturers. A novel cloud-based queuing model for TI is proposed in [15] by utilising the PD-NOMA strategy. This model uses a Baseband Processing Unit (BBU) and Radio Remote Head (RRH), queuing delays for tactile end-users. Resource allocation is formulated to reduce the latency between tactile end-users. The transmit power is reduced by choosing a dynamic approach of fronthaul and access delays rather than a fixed approach. Finally, the energy efficiency of PD-NOMA and OFDMA is compared.

In [13], the critical features of NOMA are reviewed, and it highlights the merits and demerits against other OMA strategies. The features of several NOMA schemes, such as PD-NOMA, SCMA, MUSA and Pattern Division Multiple Access (PDMA), have been discussed. In addition, the KPIs such as sum rate, energy efficiency and BER are compared between NOMA strategies. Thus, NOMA has the potential to achieve the objectives of the required KPIs. On the other hand, the authors in [16] have proposed deep-learning-based grant-free NOMA to tackle and minimise the ultra-low latency and ultra-reliability requirements in massive access scenarios. They mentioned that the combination of grant-free access and NOMA can be leveraged and is a promising approach for tactile IoT. However, random interference is caused, which lowers the system’s reliability. Hence, a grant-free NOMA-based neural network model and a novel multi-loss function are considered highly suited for automated applications in tactile IoT. Simulation results of the proposed model outperform the traditional grant-free NOMA strategies.

Moreover, in [17], the NOMA in a Mobile Edge Computing (MEC)-enabled wireless tactile IoT scenario is investigated and optimisation algorithms are derived for system and user performance. The proposed network model incorporates an MEC server at the access point, thus supporting computation for two sensor clusters. The system and cluster heads’ performance using the Successful Computation Probability (SCP) is assessed, mainly focusing on high Signal-to-Noise Ratios (SNRs) for a comprehensive understanding. The simulation results of the proposed network model outperform and boost system performance compared to traditional OMA strategies. The authors in [18] explore and analyse the application-specific NOMA-based communication infrastructure for TI, allowing non-orthogonal RB sharing among 5G generic services such as eMBB, mMTC and URLLC devices to a shared BS. A comparative analysis of NOMA and OMA is shown concerning the sum rate and number of users. Various NOMA variants are discussed to check the feasibility of future low-latency TI applications/services.

To meet the requirements of high spectral efficiency, ultra-low latency and multi-user connectivity, the authors in [8] have considered NOMA a potential solution which allows some degree of interference on the receiver side. In contrast, the OMA technique may not meet the strict requirements mentioned previously. Here, they have focussed on providing a novel NOMA model, including Uplink (UL) and Downlink (DL) transmissions in MIMO and cooperative communication scenarios. Hence, the performance of NOMA and OMA are compared to analyse the system in terms of spectral efficiency, sum rate and BER. In [19], the authors have predominantly investigated the implementation of NOMA on a Software-Defined Radio (SDR) platform. They have highlighted SDR as a flexible platform for testing and implementing 5G and B5G technologies. In addition, various SIC receivers such as Ideal SIC, Symbol-level SIC, Codeword-level SIC, and Log Likelihood Ratio (LLR)-based receivers are mathematically evaluated and analysed. When NOMA is compared with OMA with simulation results, NOMA showcases its superiority over OMA in terms of KPIs.

Furthermore, a survey on the NOMA system is presented in [20], focussing on error rate analysis. The enhanced NOMA strategies, which consist of constellation diagrams, multicarrier systems and detector designs, are discussed, along with research problems and future directions. In [21], a discussion of a deep-learning technique for the NOMA systems is presented. The emphasis is given to deep-learning-based NOMA systems for solving communication issues. These NOMA systems can be integrated with potential technologies such as MEC, MIMO, Intelligent Reflecting Surfaces (IRSs) and Simultaneous Wireless and Information Power Transfer (SWIPT). Also, the focus has been given to KPIs such as SIC, CSI, user fairness and other valuable parameters.

Considering the IRS, ref. [22] has addressed the issues related to harnessing the performance of wireless networks in 1G to 5G propagation environments. The issues can be resolved in the Sixth Generation (6G) by employing the IRS with NOMA. The designs and challenges related to IRS-based NOMA systems are comprehensively discussed, with a detailed analysis of the communication framework. In [23], a survey is conducted based on combining the benefits of NOMA and cell-free massive MIMO systems, and a detailed review of how the performance can be increased is included. Moreover, the challenges of combining cell-free massive MIMO systems with other potential technologies are discussed.

Visible light communication is a potential solution for high-speed data communications. In [24], the authors have conducted a comprehensive review of NOMA techniques with the involvement of visible light communication systems. They also discussed the limitations and challenges of integrating NOMA with visible light communication systems and the role of machine learning and physical layer security. The authors in [25] have reviewed NOMA-enabled MEC systems in depth, focusing on the issues, challenges and shortcomings that arise. They have claimed that integrating NOMA with MEC will bring umpteen performance characteristics to 5G and B5G communication infrastructures, such as energy efficiency, latency, throughput and massive end-user connectivity.

To overcome the issues related to poor channel quality and disconnected communication from BS to users, the work in [26] has analysed the BER and outage probability for multi-hop decode and forward relay-assisted NOMA systems. The BER and outage probability equations are also derived by considering imperfect SIC and CSI. The proposed model’s simulation results depict the superiority over traditional OMA systems. The authors in [9] have focussed on efficient strategies to integrate NOMA into 5G and 6G systems. This integration can be beneficial for UL and DL application environments under the 3rd Generation Partnership Project (3GPP) standardisation.

The author in [27] has focussed on optimising PA coefficients to gain optimum proportional fairness in PD-NOMA transmission with complete or limited SIC. The numerical results illustrate the impact of complete or limited SIC on the system performance with sum rate loss due to proportional fairness. Similarly, ref. [28] considers a hybrid automatic repeat request protocol for PD-NOMA with proportional fairness on the fading channel. It analyses the system performance for different symmetric and asymmetric scenarios concerning throughput, outage probabilities and delays with varying PA.

## 3. System Model

An SISO-NOMA-based power-domain multiplexing system model is considered mathematically, consisting of a BS with users ($U_{1}$, $U_{2}$, *…*, $U_{N}$). Here, we are considering a downlink SISO-NOMA communication scenario. At the transmitter side, the users’ messages are multiplexed using superposition coding, whereas, at the receiver end, the received users’ messages are demultiplexed (decoded) to retrieve the original message and perform Successive Interference Cancellation (SIC) to remove interference from other users’ messages.

### 3.1. Downlink PD SISO-NOMA Communication Scenario

Figure 2 represents the downlink power-domain communication scenario in TI. Let $m_{1}$, $m_{2}$, *…*, $m_{N}$ denote users’ messages to be transmitted from the BS, with transmitted power as $P_{T}$. Let the PA coefficients be $\alpha_{1}$, $\alpha_{2}$, *…*, $\alpha_{N}$, with the corresponding channel coefficients as $h_{1}$, $h_{2}$, *…*, $h_{N}$.

Let us consider $U_{1}$ to be the farthest user, followed by $U_{N}$ to be the nearest user to the BS.

As superposition coding is performed at the BS, the transmitted signal ($t_{s}$) from the BS is given by
(1)$$t_{s}=\sqrt{P_{T}}\left(\sqrt{\alpha_{1}}m_{1}+\sqrt{\alpha_{2}}m_{2}+\ldots +\sqrt{\alpha_{N}}m_{N}\right)$$

In short, $t_{s}$ at $i^{th}$ user can be written as
$$t_{s}=\sum_{i=1}^{N}\sqrt{P_{T}}\alpha_{i}\times m_{i}$$
(2)$$t_{s}=\sqrt{P_{T}}\sum_{i=1}^{N}\alpha_{i}\times m_{i}$$

Therefore, the received signal ($r_{s}$) at the $i^{th}$ user is given by
(3)$$r_{s}=h_{i}t_{s}+n_{i}$$

Comparing Equations (Equation 2) and (Equation 3), $r_{s}$ can be written as
$$r_{s}=h_{i}\left[\sqrt{P_{T}}\sum_{i=1}^{N}\alpha_{i}\times m_{i}\right]+n_{i}$$
(4)$$r_{s}=\sqrt{P_{T}}h_{i}\left[\sqrt{\alpha_{1}}m_{1}+\sqrt{\alpha_{2}}m_{2}+\ldots +\sqrt{\alpha_{N}}m_{N}\right]+n_{i}$$
where $n_{i}$ is Additive White Gaussian Noise (AWGN), with mean 0 and variance $\sigma^{2}$.

By considering Equation (Equation 4), the $r_{s}$ for User 1 ($U_{1}$ = farthest user), where i=1, will be
(5)$$r_{s}^{U_{1}}=\sqrt{P_{T}}h_{1}\left[\underset{Desired\;Signal}{\underset{︸}{\sqrt{\alpha_{1}}m_{1}}}+\underset{Interference\;Signal}{\underset{︸}{\sqrt{\alpha_{2}}m_{2}+\ldots +\sqrt{\alpha_{N}}m_{N}}}\right]+n_{1}$$

As User 1 is allocated with the highest power coefficients ($\alpha_{1}$), the $r_{s}$ component of User 1 is the desired and dominant signal, and other components are considered undesired (interference).

Along the same lines, $r_{s}$ for User 2 ($U_{2}$ = relatively near to BS), where i=2, will be
(6)$$r_{s}^{U_{2}}=\sqrt{P_{T}}h_{2}\left[\underset{Undesired\;Signal}{\underset{︸}{\sqrt{\alpha_{1}}m_{1}}}+\underset{Desired\;Signal}{\underset{︸}{\sqrt{\alpha_{2}}m_{2}}}+\underset{Interference\;Signal}{\underset{︸}{\ldots +\sqrt{\alpha_{N}}m_{N}}}\right]+n_{2}$$

As User 2 is allocated with a relatively lower power coefficient ($\alpha_{2}<\alpha_{1}$), the $r_{s}$ component of User 2 is considered as desired but not dominant. The dominant $r_{s}$ signal is still the User 1 signal component, which is considered undesired (interference), along with other remaining user signal components. Hence, SIC is performed on the $r_{s}$ to decode/retrieve the user’s original message/signal to remove undesired (interference) signal components.

### 3.2. SINR Analysis

To retrieve the original message from the received signal ($r_{s}$), the particular user’s received signal must be directly decoded, considering other signal components as interference.

Therefore, to decode $r_{s}$ signal for User 1 from Equation (Equation 5), the instantaneous SINR for User 1 is given as
(7)$$SINR_{U_{1}}=\frac{P_{T}\alpha_{1}{\left|h_{1}\right|}^{2}}{P_{T}\alpha_{2}|h_{1}{|}^{2}+P_{T}\alpha_{3}|h_{1}{|}^{2}+\ldots +P_{T}\alpha_{N}{\left|h_{1}\right|}^{2}+\sigma^{2}}$$
where $|h_{1}{|}^{2}$ is the channel gain for User 1.

Similarly, to decode the $r_{s}$ signal for User 2 from Equation (Equation 6), the $r_{s}$ must decode the User 1 signal and perform SIC as User 1’s received signal component is an undesired but dominant signal. Hence, after performing SIC, the resulting $r_{s}$ will be
(8)$$r_{s}^{\prime U_{2}}=\sqrt{P_{T}}h_{2}\left[\underset{Desired\;Signal}{\underset{︸}{\sqrt{\alpha_{2}}m_{2}}}+\underset{Interference\;Signal}{\underset{︸}{\ldots +\sqrt{\alpha_{N}}m_{N}}}\right]+n_{2}$$

Once User 1’s dominant signal component is removed from Equation (Equation 6), resulting in Equation (Equation 8), the $r_{s}$ signal can be directly decoded so that User 2 can have the desired and dominant signal after SIC. Hence, other remaining user signal components can be treated as interference. Hence, the instantaneous SINR for User 2 is given as
(9)$$SINR_{U_{2}}=\frac{P_{T}\alpha_{2}{\left|h_{2}\right|}^{2}}{P_{T}\alpha_{3}|h_{2}{|}^{2}+\ldots +P_{T}\alpha_{N}{\left|h_{2}\right|}^{2}+\sigma^{2}}$$
where $|h_{2}{|}^{2}$ is the channel gain for User 2.

Therefore, the SINR for the $i^{th}$ user can be expressed as
$$SINR_{U_{i}}=\frac{P_{T}\alpha_{i}{\left|h_{i}\right|}^{2}}{P_{T}\alpha_{i+1}|h_{i}{|}^{2}+\ldots +P_{T}\alpha_{N}{\left|h_{i}\right|}^{2}+\sigma^{2}}$$
(10)$$SINR_{U_{i}}=\frac{P_{T}\alpha_{i}{\left|h_{i}\right|}^{2}}{\sum_{j=i+1}^{N}P_{T}\alpha_{j}{\left|h_{i}\right|}^{2}+\sigma^{2}}$$
where $|h_{i}{|}^{2}$ is the channel gain for the $i^{th}$ user.

### 3.3. Sum Rate Analysis

The achievable sum rate of the $i^{th}$ user for downlink SISO-NOMA can be computed as
(11)$$R_{U_{i}}=\log_{2}\left(1+SINR_{U_{i}}\right)$$

From Equations (Equation 10) and (Equation 11), $R_{U_{i}}$ can be written as
(12)$$R_{U_{i}}=\log_{2}\left(1+\frac{P_{T}\alpha_{i}{\left|h_{i}\right|}^{2}}{\sum_{j=i+1}^{N}P_{T}\alpha_{j}{\left|h_{i}\right|}^{2}+\sigma^{2}}\right)$$

Therefore, the overall achievable sum rate for the SISO-NOMA downlink can be expressed as,
(13)$$\begin{array}{c} \begin{array}{cc} R_{overall} & =\sum_{i=1}^{N}R_{U_{i}}\\ \; & =\sum_{i=1}^{N}\log_{2}\left(1+\frac{P_{T}\alpha_{i}{\left|h_{i}\right|}^{2}}{\sum_{j=i+1}^{N}P_{T}\alpha_{j}{\left|h_{i}\right|}^{2}+\sigma^{2}}\right)\\ \; & =\sum_{i=1}^{N-1}\log_{2}\left(1+\frac{P_{T}\alpha_{i}{\left|h_{i}\right|}^{2}}{\sum_{j=i+1}^{N}P_{T}\alpha_{j}{\left|h_{i}\right|}^{2}+\sigma^{2}}\right)\\ \; & \;\;\;\;+\log_{2}\left(1+\frac{P_{T}\alpha_{N}{\left|h_{N}\right|}^{2}}{\sigma^{2}}\right) \end{array} \end{array}$$

To evaluate the SISO-NOMA scheme at higher SNR, the variance $\sigma^{2}$ tends to 0 ($\sigma^{2}\to 0$). Hence, from Equation (Equation 13), the achievable sum rate can be approximated as
$$\begin{array}{c} \begin{array}{cc} R_{overall} & \approx \sum_{i=1}^{N-1}\log_{2}\left(1+\frac{P_{T}\alpha_{i}{\left|h_{i}\right|}^{2}}{\sum_{j=i+1}^{N}P_{T}\alpha_{j}{\left|h_{i}\right|}^{2}+\sigma^{2}}\right)\\ \; & \;\;\;\;+\log_{2}\left(1+\frac{P_{T}{\left|h_{N}\right|}^{2}}{\sigma^{2}}\right) \end{array} \end{array}$$
(14)$$R_{overall}\approx \log_{2}\left(\frac{P_{T}{\left|h_{N}\right|}^{2}}{\sigma^{2}}\right)$$

### 3.4. Fair PA Analysis

The fair PA in SISO-NOMA downlink communication is crucial to ensure that all system users ($U_{1}$, $U_{2}$, *…*, $U_{N}$) have a guaranteed QoS and to maximise the sum rate. Thus, fair PA promotes user fairness, balances interference (undesired signal components) and enhances spectral efficiency.

Moreover, the PA coefficients ($\alpha_{1}$, $\alpha_{2}$, *…*, $\alpha_{N}$) are directly dependent on channel conditions. Without considering the channel conditions, the fixed PA coefficients are used, where the outage probabilities of the users are higher with a lower sum rate (bps/Hz).

Considering the CSI, the PA coefficients can be fairly optimised to improve the system performance. The fair PA focuses on the far user (User 1 in our case), as the far user is weak and located relatively far away from the BS compared to other users.

To better evaluate the SISO-NOMA scheme, let us consider two users (User 1—far user, and User 2—near user) presented in the system. From Equation (Equation 12), the achievable sum rate for Users 1 and 2, respectively, can be given as
(15)$$R_{U_{1}}=\log_{2}\left(1+\frac{P_{T}\alpha_{1}{\left|h_{1}\right|}^{2}}{P_{T}\alpha_{2}{\left|h_{1}\right|}^{2}+\sigma^{2}}\right)$$
(16)$$R_{U_{2}}=\log_{2}\left(1+\frac{P_{T}\alpha_{2}{\left|h_{2}\right|}^{2}}{\sigma^{2}}\right)$$

Hence, in this case, the PA coefficients ($\alpha_{1}$ and $\alpha_{2}$) are designed/derived by setting up the target rate ($R_{T}$) for the far user (User 1), which is less than or equal to the sum rate, i.e., $R_{T}\leq R_{U_{1}}$, from Equation (Equation 15). Once the $R_{T}$ for User 1 is met, the optimised PA coefficients can be derived instead of having a fixed PA.

Therefore, Equation (Equation 15) becomes
(17)$$R_{T}=\log_{2}\left(1+\frac{P_{T}\alpha_{1}{\left|h_{1}\right|}^{2}}{P_{T}\alpha_{2}{\left|h_{1}\right|}^{2}+\sigma^{2}}\right)$$
$$2^{R_{T}}-1=\frac{P_{T}\alpha_{1}{\left|h_{1}\right|}^{2}}{P_{T}\alpha_{2}{\left|h_{1}\right|}^{2}+\sigma^{2}}$$

Let us consider $2^{R_{T}}-1=\beta$
$$\beta =\frac{P_{T}\alpha_{1}{\left|h_{1}\right|}^{2}}{P_{T}\alpha_{2}{\left|h_{1}\right|}^{2}+\sigma^{2}}$$
(18)$$\beta P_{T}\alpha_{2}|h_{1}{|}^{2}+\beta \sigma^{2}=P_{T}\alpha_{1}{\left|h_{1}\right|}^{2}$$

We know that the sum of the PA coefficients is equal to 1 as we have two users (User 1 and User 2, having PA coefficients of $\alpha_{1}$ and $\alpha_{2}$, respectively), $\alpha_{1}+\alpha_{2}=1$. Therefore, Equation (Equation 18) implies
(19)$$\alpha_{1}=\frac{\beta (P_{T}|h_{1}{|}^{2})+\sigma^{2}}{\left(1+\beta \right)P_{T}{\left|h_{1}\right|}^{2}};\alpha_{1}<1$$

Hence, Equation (Equation 19) can be also written as
$$\alpha_{1}=\min\left(\frac{\beta (P_{T}|h_{1}{|}^{2})+\sigma^{2}}{\left(1+\beta \right)P_{T}{\left|h_{1}\right|}^{2}},1\right)$$

Once the PA coefficient ($\alpha_{1}$) of a far user (User 1) is calculated, the PA coefficient ($\alpha_{2}$) of a near user (User 2) can be calculated using
(20)$$\alpha_{2}=1-\alpha_{1}$$

In this case, if the calculation for the term $\left(\frac{\beta (P_{T}|h_{1}{|}^{2})+\sigma^{2}}{\left(1+\beta \right)P_{T}{\left|h_{1}\right|}^{2}}\right)>1$ (let us consider 20), then $\alpha_{1}=\min\left(\frac{\beta (P_{T}|h_{1}{|}^{2})+\sigma^{2}}{\left(1+\beta \right)P_{T}{\left|h_{1}\right|}^{2}},1\right)=1$, which does not satisfy the condition $\alpha_{1}+\alpha_{2}=1$.

If $\alpha_{1}$ is designed to have 20, then User 1 will meet the target rate ($R_{T}$). But, if $\alpha_{1}<20$, User 1 will not meet $R_{T}$, leading to an outage condition. Moreover, $\alpha_{1}$ cannot hold the value 20, which violates the condition $\alpha_{1}+\alpha_{2}=1$. Consequently, even if User 1’s $\alpha_{1}$ holds the value 1 (not meeting the considered condition $\alpha_{1}=20$), User 1 will be in an outage situation.

In contrast, if $\alpha_{1}=1$, then $\alpha_{2}=1-\alpha_{1}$ will be 0. This infers that User 2 will not have any PA coefficient. Hence, User 2 will also be in an outage condition.

The solution for this outage problem of Users 1 and 2 can be resolved by setting up $\alpha_{1}$ as 0 (no PA) and $\alpha_{2}$ as 1 (entire PA). To keep User 1 away from the outage situation, the considered value of $\alpha_{1}$ must be kept at 20. But the ideal value of $\alpha_{1}$ cannot exceed 1, i.e., for $\alpha_{1}<20$; User 1 will still be in an outage and not meet the desired target rate. Therefore, allocating the entire PA to User 2 ($\alpha_{2}=1$) can be considered to a point where allocating any PA to User 1 will not matter or affect the outage condition, i.e., User 1 will be in outage condition even if PA is done. Thus, User 2 will not be in the outage, achieving a higher sum rate (bps/Hz).

### 3.5. Beamforming with Scheduling Process for 4 × 4 MIMO Use-Case Scenario Analysis

For the downlink PD-NOMA scenario, an SISO-based NOMA has been considered and discussed in the previous sections, which is extended for the MIMO-based NOMA scenario.

In this scenario, consider a use-case having a BS with a Uniform Linear Array (ULA) of 4 antennas with half-wavelength spacing. The BS serves 5 clusters/pairs of a total of 10 legitimate users, with 2 users in each cluster. Each user is equipped with 4 receiving antennas. Assume the users in each cluster have almost the same angles with clusters equally spaced between −60° to +60°, i.e., −60°, −30°, 0°, +30° and +60°, concerning ULA.

In our multi-user 4 × 4 MIMO-NOMA scenarios, the beamforming technique proposed in [29,30,31], especially Zero Forcing-based Beamforming (ZF-BF), can be used to mitigate the interference caused by multiple clusters of users, where multiple users’ clusters are simultaneously served within the same RB. Such interference can lead to performance reduction and signal quality degradation. In ZF-BF, the channel matrix is formed to represent a BS and clusters of users’ channel conditions and is used to calculate the precoding matrix by pseudo-inverting the channel matrix. Each column of the precoding matrix represents the beamforming vectors. So, the beamforming vectors are designed in such a way that they carefully direct/beam/steer the transmitted signal spatially to the desired/targeted users, forcing nulls to undesired/untargeted users or users’ clusters, thus without causing interference between them. In this scenario, having 5 clusters, ZF-BF can serve 2 clusters (with a total of 4 users) simultaneously, thus ensuring users within a cluster and inter-cluster do not interfere among themselves. Hence, ZF-BF helps improve the achievable sum rate, resource management and signal quality.

Moreover, the scheduling process proposed in [32,33], specifically the round-robin scheduling process, can be utilised by a BS to serve clusters of users simultaneously during each time slot. After serving a cluster of users, the BS selects the next cluster of users in a round-robin sequence, thus managing equal access for each cluster to channel over time and promoting user fairness. Hence, in our case, with 4 × 4 MIMO-NOMA, 2 clusters with 2 users each (with a total of 4 users) can actively acquire channel access at a given time slot. The remaining cluster users will get their turn for the channel evenly over the following time slots. Imperfect CSI and SIC are not taken into consideration in this scenario.

It is worth noting that joint dynamic user clustering, beamforming and a scheduling process can be mathematically modelled for MIMO-NOMA in a tactile communication infrastructure, which is beyond the paper’s scope and will be included in our future work.

### 3.6. Latency Analysis

In the downlink power-domain communication scenario in TI, latency can be defined as the delay incurred when the signal (ts) is transmitted from the BS until it is received by users ($U_{1}$, $U_{2}$, *…*, $U_{N}$). This latency includes delays such as transmission delay, propagation delay, queuing delay and processing delay.

The transmission delay ($D_{T}$) is the time required to transmit data packets over the communication channel. This delay considers the sum rate (Equations (Equation 15) and (Equation 16)) or set target rate (Equation (Equation 17)) of users and the amount of data to be sent (packet size). Thus, $D_{T}$ is given by
(21)$$D_{T}=\frac{Packet\;size\;in\;bits}{Transmission\;rate\;in\;bps}$$
where transmission rate = sum rate in (bps/Hz) × system bandwidth (Hz).

Moreover, the propagation delay ($D_{P}$) is the time required for the data packets to travel the physical distance between the BS and the user. This delay also relies on the transmission rate of the signal and thus is given by
(22)$$D_{P}=\frac{Distance\;between\;the\;BS\;and\;the\;user\;in\;metres}{Speed\;of\;light\;in\;m/s}$$
where speed of light = $3\times 10^{8}$ m/s.

The queuing delay ($D_{Q}$) is the time required to wait in queues at the BS before data packet transmission. This delay may arise due to network congestion or scheduled multiple users based on channel conditions. On the other hand, the processing delay ($D_{Pr}$) is the time required to process superposition coding (multiplexing) at the BS and SIC decoding at the receiver’s side.

Hence, the overall latency (L_overall) for a particular user will be given by
(23)$$L\text{\_}overall=D_{T}+D_{P}+D_{Q}+D_{Pr}$$

## 4. Performance Evaluation

### 4.1. Simulation Environment

To evaluate the performance of the NOMA system model, the MATLAB 2024a [34] platform is used along the 5G communication toolbox and required packages to carry out the simulations. This platform models the transmitter (BS), receiver (users) and channel condition of the NOMA system. Particularly, it can also model fading channels, such as Rayleigh fading, and simulate the path loss to mimic real-world propagation conditions.

Moreover, Monte Carlo simulation is applied to generate Rayleigh fading channels using random complex Gaussian variables, add AWGN to received signals at the receiver end and run multiple simulation runs to achieve statistically meaningful numerical results.

### 4.2. Simulation Results and Discussion

To critically compare and analyse our proposed SISO-NOMA system in TI, we observe and measure KPIs, such as Bit Error Rate (BER), achievable sum rate and outage probability. The fixed PA coefficient pairs are used to calculate the trend of the mentioned KPIs. Furthermore, to infer meaningful insights, an analysis of performance comparisons for BER and achievable sum rates is made between SISO-NOMA and OMA. In addition, the effect of fair PA on certain users is also considered whilst being compared against the fixed PA to see the outage probability and achievable sum rate trends of the users. Moreover, a comparative latency analysis is conducted for SISO-NOMA and OMA with fixed and fair PAs. Finally, the performance comparison concerning achievable sum rate and latency is analysed between 4 × 4 MIMO-NOMA and SISO-NOMA. The simulation results reported in this paper showed steady-state behaviour with a relative statistical error of ≤5% at a 95% confidence level. Table 1 lists the simulation parameters used in the proposed SISO-NOMA-based TI system.

#### 4.2.1. Performance Comparison and Analysis of Bit Error Rate (BER) Between SISO-NOMA and OMA

To study the effect of varying path loss exponent (η) with fixed PA coefficient values, the performance comparison of BER is observed and measured for the SISO-NOMA and OMA schemes using two-user and three-user scenarios, with the following cases: *a*, *b*, *c* and *d*. The PA coefficients for Users 1, 2 and 3 are denoted as $\alpha_{1}$, $\alpha_{2}$ and $\alpha_{3}$, respectively. The varying η allows system behaviour analysis under different environmental conditions. When η increases, the system represents the received signal’s faster decay with distance.

Case (*a*): Considering a two-user scenario with η=2 and fixed PA coefficient pairs as ($\alpha_{1}=0.70$ & $\alpha_{2}=0.30$) and ($\alpha_{1}=0.80$ & $\alpha_{2}=0.20$).

The BER performance trend is observed in Figure 3a against a varying Signal-to-Noise Ratio (SNR) in decibels (dB) for the SISO-NOMA and OMA schemes, having a two-user scenario with η=2 and fixed PA coefficient pairs as ($\alpha_{1}=0.70$ & $\alpha_{2}=0.30$) and ($\alpha_{1}=0.80$ & $\alpha_{2}=0.20$) for Users 1 and 2. User 1 (weak user) is located far away from the BS, whereas User 2 (strong user) is located near the BS. It is observed that User 1, with $\alpha_{1}$ having 70%, has incurred more BER than User 1, with $\alpha_{1}$ having 80%, at an SNR = 80 dB. This performance trend is because more power has been allocated to User 1, with 80%, than User 1, with 70%. In contrast, even though higher power is allocated to User 2, with 30%, it shows a higher BER than User 2, with 20%. This is mainly due to cross-user interference between User 1, with 70%, and User 2, with 30%. Thus, there is less interference between Users 1, with 80%, and 2, with 20%. Keeping η=2 constant, when the BER performance trend with the SISO-NOMA scheme is compared with the OMA scheme, it is seen that lower BER is noted for the user pairs (with 70% and 30%) and (with 80% and 20%). In the SISO-NOMA scheme, the extra signal processing complexity is added at the receiver end, i.e., decoding and performing SIC of received signals, thus leading to higher BER than in the OMA scheme.

Case (*b*): Considering a two-user scenario with η=4 and fixed PA coefficient pairs as ($\alpha_{1}=0.70$ & $\alpha_{2}=0.30$) and ($\alpha_{1}=0.80$ & $\alpha_{2}=0.20$).

On similar lines, the BER performance trend is observed in Figure 3b with the same fixed PA coefficient pairs as ($\alpha_{1}=0.70$ & $\alpha_{2}=0.30$) and ($\alpha_{1}=0.80$ & $\alpha_{2}=0.20$) for Users 1 and 2, but changing η from 2 to 4. For this simulation case, the effect of increasing η is noted for BER performance for Users 1 and 2. In Figure 3b, we observe that a slightly lower BER is encountered for User 1, with 80%, than for User 1, with 70%, at SNR = 150 dB. In contrast, a similar trend is observed for Case (a), followed by User 2, which has a lower BER of 20%, than User 2, which has 30%. Moreover, in the OMA scheme, smaller BER is observed than in the proposed SISO-NOMA scheme. Considering a constant η=4, as the SNR increases BER tends to improve as the received signal becomes more distinguishable than the noise signal. Hence, improvement in BER performance is observed when η is increased from 2 to 4 and the PA coefficient pairs are kept fixed. Consequently, a trade-off is observed in BER performance to strike a balance between η and PA coefficients to optimise the communication performance.

Case (*c*): Considering a three-user scenario with η=2 and fixed PA 3-tuple coefficients as ($\alpha_{1}=0.70$, $\alpha_{2}=0.20$ & $\alpha_{3}=0.10$) and ($\alpha_{1}=0.76$, $\alpha_{2}=0.16$ & $\alpha_{3}=0.08$).

The BER performance trend is observed in Figure 4a against varying Signal-to-Noise Ratio (SNR) in decibels (dB) for the SISO-NOMA and OMA schemes having a three-user scenario with η=2 and fixed PA 3-tuple coefficients as ($\alpha_{1}=0.70$, $\alpha_{2}=0.20$ & $\alpha_{3}=0.10$) and ($\alpha_{1}=0.76$, $\alpha_{2}=0.16$ & $\alpha_{3}=0.08$) for Users 1, 2 and 3. In this three-user scenario, User 1 is located far away from the BS, whereas User 3 is located near the BS. User 2 is located in the middle of User 1 and User 3. We observe that User 1, with 70%, has incurred higher BER than User 1, with 76%, at SNR = 90 dB. This is because less power (70%) is allocated to User 1 than to User 1, with 76%. Contrarily, lower BER is observed for User 2, with 16%, than User 2, with 20%, even though higher power is allocated to User 2, with 20%. In both cases, cross-user interference is observed between the User 2 PA coefficients, which are 16% and 20%, respectively. For User 3 (with 10% and 8%), almost the same BER is noted. As η=2 is also kept constant for OMA scheme users, it can be clearly perceived that the BER performance trend is relatively less than SISO-NOMA scheme users, as OMA users do not have to undergo decoding and SIC operations for received signals.

Case (*d*): Considering a three-user scenario with η=4 and fixed PA 3-tuple coefficients as ($\alpha_{1}=0.70$, $\alpha_{2}=0.20$ & $\alpha_{3}=0.10$) and ($\alpha_{1}=0.76$, $\alpha_{2}=0.16$ & $\alpha_{3}=0.08$).

On the same note, Figure 4b shows the BER performance trend for η=4 (instead of 2) and the same fixed PA 3-tuple coefficients as ($\alpha_{1}=0.70$, $\alpha_{2}=0.20$ & $\alpha_{3}=0.10$) and ($\alpha_{1}=0.76$, $\alpha_{2}=0.16$ & $\alpha_{3}=0.08$). The same BER performance is observed for Case (*c*) with User 1 (with 70% and 76%) at an SNR = 140 dB. However, User 2, with 20%, has shown higher BER than User 2, with 16%. User 2, with 20%, has experienced more cross-user interference with User 1, with 70%, while User, 2, with 16%, has experienced less cross-user interference with User 1, with 76%. Approximately the same BER is observed for User 3 (with 10% and 8%). When OMA scheme users are compared to SISO-NOMA users, lower BER is noted for far, middle and near users with higher SNR values. It can be inferred that when η is increased from 2 to 4 in a three-user scenario, the degradation (increment) of BER performance is observed with η=4 at higher SNR when compared with η=2 due to increased interference.

#### 4.2.2. Performance Comparison and Analysis of Achievable Sum Rate Between SISO-NOMA and OMA

Figure 5 shows the achievable sum rate in bps/Hz with respect to SNR values for SISO-NOMA and OMA. Considering a three-user scenario, Users 1, 2 and 3 have been allocated with fixed PA coefficients as $\alpha_{1}=0.70$ (70%), $\alpha_{2}=0.20$ (20%) and $\alpha_{3}=0.10$ (10%) and path loss exponent as η=2. It can be observed that the achievable sum rate in the OMA performs slightly better than the SISO-NOMA at low SNR values. Due to cross-user interference and simulation transmission using the same RB, the achievable sum rate in the SISO-NOMA scheme suffers when compared to the OMA scheme at low SNR values.

However, as SNR increases, the achievable sum rate in the proposed SISO-NOMA scheme outperforms the OMA scheme.

#### 4.2.3. Performance Analysis of Outage Probability in SISO-NOMA

The performance analysis of outage probability in the SISO-NOMA scheme is carried out using a two-user scenario with fixed PA coefficient pairs as ($\alpha_{1}=0.70$ & $\alpha_{2}=0.30$) and ($\alpha_{1}=0.80$ & $\alpha_{2}=0.20$) and path loss exponent η=4. To plot the outage probabilities, let us consider the target rates for Users 1 and 2 as 1.5 bps/Hz and 2.5 bps/Hz. These considered target rates are now compared with the achievable sum rate calculated in Equations (Equation 15) and (Equation 16). If the calculated achievable sum rates drop below the respective considered target rates of Users 1 and 2, the individual counters will be incremented. Hence, the outage probability is plotted as a function of power transmitted for both Users 1 and 2.

From Figure 6, it can be observed that User 1, with 70%, has more outage probability than User 1, with 80%. On the other hand, User 2, with 30%, has a lower outage probability than User 2, with 20%. Therefore, it can be concluded that the outage probability depends on the power percentage allocated to the particular user. Thus, in this case, User 1, with 80%, and User 2, with 30%, have shown less outage probability than User 1, with 70%, and User 2, with 20%, respectively.

#### 4.2.4. Performance Comparison and Analysis of Fair PA with Fixed PA

The outage probability with fair PA and fixed PA are plotted against the target rate in a two-user scenario with η=4 (Figure 7a,b). For this simulation, the achievable sum rate derived in Equation (Equation 15) for User 1 (far from BS) is considered the same as the target rate for User 1 (far). The PA coefficients $\alpha_{1}$ and $\alpha_{2}$ are calculated using derived Equations (Equation 19) and (Equation 20) for fair PA. The fixed PA coefficients for Users 1 and 2 are used as $\alpha_{1}=0.80$ and $\alpha_{2}=0.20$, respectively (for the transmitted power of 40 dBm).

By looking at Figure 7a, one can observe that Users 1 and 2 in fixed PA coefficients are in outage condition when the target rate is greater than 2 bps/Hz. This means that Users 1 and 2 in fixed PA will not experience an outage if the target rate is less than 2 bps/Hz. To infer, users in fixed PA are poorly performing and saturated to 1, as fixed PA neither considers the user’s target rate requirements nor utilises instantaneous CSI conditions. Hence, it is not an optimal strategy to allocate power to users in the network. On the other hand, for a fair PA case, coefficients $\alpha_{1}$ and $\alpha_{2}$ are derived by considering target rate requirements dependent on the achievable sum rate of User 1 (far), i.e., CSI. As the target rate increases, User 1 (far) gradually goes into an outage condition, thus increasing the outage probability. However, User 2 (near) shows a sudden change in the outage probability trend when the target rate is between 5 bps/Hz and 8.5 bps/Hz. After 8.5 bps/Hz, User 2 also goes into an outage condition.

Figure 7b shows the improved outage probability for a fair PA case for the required target rate. The improved outage probability trend is achieved by forcefully setting User 1 (far) and User 2 (near) PA coefficients as $\alpha_{1}=0$ and $\alpha_{2}=1$. This setting is done when it does not affect the outage condition of User 1. When the target rate is less than 8.5 bps/Hz, derived PA coefficient $\alpha_{1}$ is assigned to User 1 (far) without bothering about User 2 (near), thus focussing more on User 1’s performance. However, when the target rate is above 8.5 bps/Hz, the PA coefficient $\alpha_{2}=1$ is assigned to User 2 (near), thus allocating the whole power to it and focussing on User 2’s performance. Hence, User 2’s outage probability is improved from 8.5 bps/Hz. User 1 has left no power ($\alpha_{1}=0$), which does not affect its outage condition.

Figure 8 shows that the achievable sum rate for having a fair PA is higher than for having a fixed PA when plotted against transmitted power. Because of varying CSI conditions, the fair PA coefficients $\alpha_{1}$ and $\alpha_{2}$ are calculated to meet the desired target rate requirements, thus achieving an improved achievable sum rate in a fair PA case.

#### 4.2.5. Performance Comparison and Analysis of Latency

The performance comparison in terms of latency is conducted and observed with a fixed η in the following cases: *a* and *b*. In our simulation, the transmission and processing delays are the major contributors to the latency for the users, whereas queuing and propagation delays are considered negligible. For the queuing delay to be negligible, the simulation does not consider multiple users sending traffic simultaneously. Considering 2- and 3-user scenarios, the likelihood of traffic congestion is quite low without queueing the packets on the BS side, thus creating less traffic demand at the BS. Since the propagation delay depends on the distance between the BS and the user, the simulation scenario considers user distances from the BS, such as 1000 m, 500 m and 200 m. Hence, the propagation delay will be in the order of μs, thus treating it as negligible.

Case (*a*): Considering a three-user scenario and comparing the latency trend for the SISO-NOMA and OMA schemes.

The latency trend is observed in Figure 9 as a function of SNR for the SISO-NOMA and OMA schemes having a three-user scenario with η=2 and fixed PA coefficients ($\alpha_{1}=0.70$, $\alpha_{2}=0.20$ & $\alpha_{3}=0.10$). The latency is plotted for each user in both cases, i.e., SISO-NOMA and OMA. This figure represents how the latency varied with varying SNR (signal quality) having three different user distances from the BS. We observe that the latency decreases with increasing SNR. At low SNR, the latencies in SISO-NOMA show higher latency than in OMA as the system consumes more time to multiplex (superposition coding) and decode (SIC) the signal. However, as the signal improves with increasing SNR and negligible noise, the rapid reduction of latencies in SISO-NOMA is observed around 5 dB–10 dB due to appropriate PA and simultaneous access to RB. As User 1 is located farthest from the BS and most likely goes to an outage condition, more power (70%) is allocated to ensure efficient sharing of resources, thus contributing to the higher latency than User 2 (20%) and User 3 (10%) in SISO-NOMA.

The signal quality is poor for OMA at low SNR, leading to higher BER, as observed in Figure 4a. Hence, frequent retransmissions are required, increasing each user’s latencies. Figure 9 shows that, at low SNR, the latencies for each user are higher. However, as SNR improves, the decreasing trend of latencies for each user is observed as SNR increases from 15 dB. Moreover, the latencies depend on users’ respective distances from the BS, as a similar trend is observed for SISO-NOMA users’ latency. To compare the performance in terms of latency, SISO-NOMA outperforms OMA as SNR increases.

Case (*b*): Considering a two-user scenario and comparing the latency trend for fair and fixed PAs in SISO-NOMA.

Figure 10 represents the latency trend for fair (derived in Equations (Equation 19) and (Equation 20)) and fixed ($\alpha_{1}=0.80$ and $\alpha_{2}=0.20$) PAs as a function of the transmitted power in SISO-NOMA. The latency trends for fair and fixed PA cases decrease with increasing transmitted power. However, the decrease in latency for fair PA is more pronounced than in fixed PA, as fair PA dynamically adjusts the power allocated to SISO-NOMA users based on channel conditions. Such fair PA adjustments most likely provide better latency performance because of the efficient utilisation of resources. On the contrary, for fixed PA, users are allocated a fixed proportion of power independent of channel conditions, thus leading to inefficient use of the available power. In comparison, better latency performance is observed in fair PA than in fixed PA as the transmitted power increases.

#### 4.2.6. Performance Comparison and Analysis of 4 × 4 MIMO Scenario in NOMA

In our simulation, the performance comparison of 4 × 4 MIMO-NOMA with SISO-NOMA is conducted and observed in terms of the achievable sum rate and latency with a fixed η=4 and 10 users, as previously mentioned in Section 3.5. The simulation scenario considers the user’s distances to be between 1000 m and 200 m from a BS, with User 1 being farthest and User 10 being closest to the BS. From User 1 to User 10, the distances from the BS are 1000, 900, 800, 700, 600, 500, 400, 300, 250 and 200 metres. Five user clusters are formed, with 2 users in each cluster with a near–near and far–far clustering approach. The achievable sum rate and latency comparison are analysed in the following cases: *a* and *b*.

Case (*a*): Performance comparison and analysis of achievable sum rate between 4 × 4 MIMO-NOMA and SISO-NOMA.

The achievable sum rate comparison is observed for 4 × 4 MIMO-NOMA and SISO-NOMA in Figure 11 when plotted against transmitted power. This figure represents an increase in the achievable sum rate as the transmit power increases for both cases, 4 × 4 MIMO-NOMA and SISO-NOMA, thus improving SNR for higher transmitted power. It can be clearly seen that 4 × 4 MIMO-NOMA outperforms SISO-NOMA as transmitted power increases. This is because of the exploiting of spatial diversity and multiplexing, thus allowing multiple data streams to be transmitted from the BS to multiple users simultaneously in 4 × 4 MIMO-NOMA. Moreover, the incorporation of ZF-BF further improves the performance of 4 × 4 MIMO-NOMA by reducing the spatial inter-user and inter-cluster interference. In contrast, a round-robin scheduling process promotes fair user access to the RB as it cycles through the clusters of users and allocates time slots sequentially, which does not prevent any clusters from having access to RB and does serve each cluster in each round.

Case (*b*): Performance comparison and analysis of latency between 4 × 4 MIMO-NOMA and SISO-NOMA.

Figure 12 represents the latency plot of 4 × 4 MIMO-NOMA and SISO-NOMA against the increasing transmitted power. It can be observed that 4 × 4 MIMO-NOMA incurs less latency than SISO-NOMA. This is because of effectively nullifying interference in ZF-BF; thus, clusters of users do not have to wait for a clear transmission slot from the BS and are served simultaneously. Also, it helps in interference-free streams for each cluster of users, thus reducing the need for retransmission and error correction and contributing to overall lower latency. However, the complexity of designing a beamforming matrix for many antennas or users may potentially increase the overall latency. However, with careful beamforming matrix design, reduced interference and parallel data transmissions offset latency. On the other hand, all clusters of users are served sequentially in a round-robin scheduling process, thus reducing the waiting time of clusters of users, promoting user fairness and overall latency. Also, this scheduling process reduces the queuing delay as it cycles through the clusters of users. Without any scheduling process, the clusters of users may experience delays or have to wait longer for their turn for transmission, thus making higher overall latency and lesser system responsiveness and avoiding constant allocation of RB to high-priority users or users having channel conditions.

### 4.3. Model Assessment and Validation

This section assesses and validates the robustness of our system model by comparing it with existing work done in incorporating NOMA in TI wireless networks.

To the best of our author’s knowledge, no existing works have focussed and analysed mathematically derived SINR, achievable sum rate and fair PA coefficients altogether to meet the TI user’s stringent requirements. Most of the existing work done in [8,12,16,20,26] have focused on and shared primitive analyses on downlink PD SISO-NOMA, considering mostly two users in a scenario. The NOMA performance has not been rigorously evaluated and analysed on varying network parameters, such as PA coefficients and η, to observe users’ performance effects. Moreover, the comparison between SISO-NOMA and traditional OMA concerning BER, sum rate and latency trends with varying parameters mentioned earlier with a two-user and a three-user scenario has not been well studied in the networking literature.

Nonetheless, the fair PA analysis in terms of outage probability presented in our work has focussed on deriving PA coefficients to meet the stringent target rates of users under TI communication. Therefore, the PA coefficients can be precisely calculated and tuned to meet a specific SISO-NOMA user’s target rate requirement. In addition, the achievable sum rate and latency with calculated PA coefficients are compared with fixed PA coefficients, which was also missing from previous works. Along similar lines, the performance of 4 × 4 MIMO-NOMA incorporating zero forcing-based beamforming and a round-robin scheduling process are compared with SISO-NOMA in terms of achievable sum rate and latency, which is not exclusively mentioned in previous work.

However, this research did not focus on machine intelligence with imperfect SIC conditions, which can be considered for future research directions.

## 5. Conclusions

A novel downlink PD SISO-NOMA communication scenario for TI employing multiple sensors and actuators, collectively treated as users, and a base station is proposed in this paper. We have developed an analytical system model comprising SINR, achievable sum rate, fair power allocation (PA) coefficients and latency for SISO-NOMA users to study the system performance. The system model is validated by simulation scenarios with varying path loss exponents and fixed PA coefficients. We have evaluated and validated the analytical model by simulation. A higher BER is achieved for SISO-NOMA users than for the OMA scheme due to additional signal processing to decode and perform Successive Interference Cancellations (SICs) for received signals. In contrast, the achievable sum rate and latency trends for the proposed PD SISO-NOMA have outperformed the OMA scheme for higher SNR. The outage probability of SISO-NOMA is also analysed with varying fixed PA coefficients. Finally, to promote dynamic PA and user fairness of the proposed PD SISO-NOMA scheme, we have compared the outage probability, achievable sum rate and latency for fixed and derived fair PA coefficients. Thus, it maximised the spectral efficiency while minimising the power consumption and latency by efficiently utilising the available spectrum/resources. Incorporating zero forcing-based beamforming and a round-robin scheduling process in 4 × 4 MIMO-NOMA has outperformed SISO-NOMA in terms of achievable sum rate and latency. Proposing a joint dynamic user clustering, beamforming and scheduling process for MIMO-NOMA and developing a deep-learning-based NOMA algorithm are suggested as future research work.

## Figures and Tables

**Figure 1 sensors-24-07220-f001:**
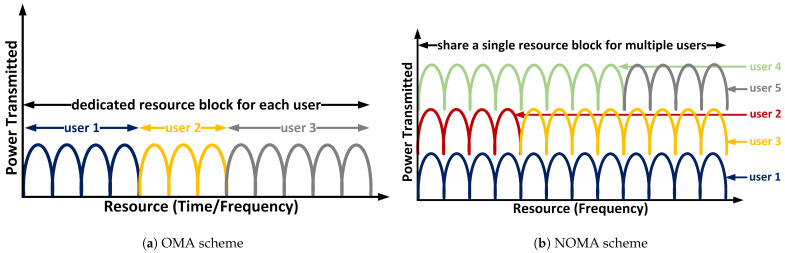
Illustrating of OMA and NOMA schemes.

**Figure 2 sensors-24-07220-f002:**
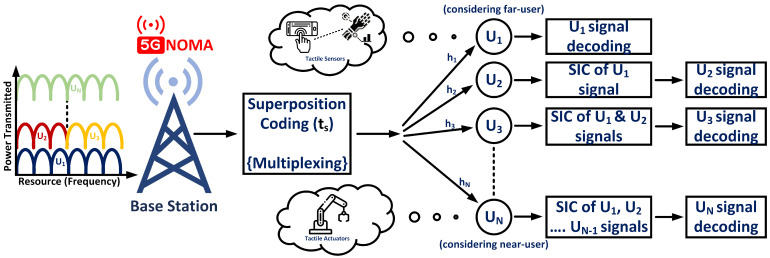
Downlink power-domain communication scenario in TI.

**Figure 3 sensors-24-07220-f003:**
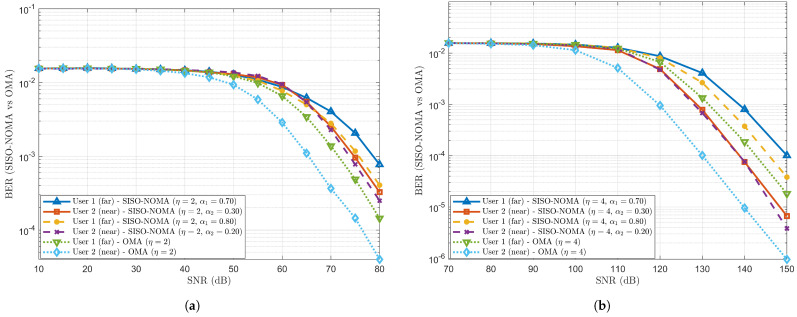
BER comparison between SISO-NOMA and OMA with η = 2 and 4, and fixed PA coefficient pairs as ($\alpha_{1}=0.70$ and $\alpha_{2}=0.30$) and ($\alpha_{1}=0.80$ and $\alpha_{2}=0.20$). (**a**) BER comparison between SISO-NOMA and OMA with η as 2. (**b**) BER comparison between SISO-NOMA and OMA with η as 4.

**Figure 4 sensors-24-07220-f004:**
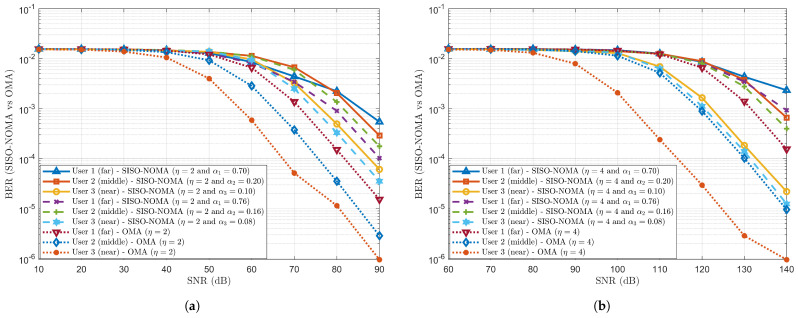
BER comparison between SISO-NOMA and OMA with η = 2 and 4, and fixed PA coefficient pairs as ($\alpha_{1}=0.70$, $\alpha_{2}=0.20$ and $\alpha_{3}=0.10$) and ($\alpha_{1}=0.76$, $\alpha_{2}=0.16$ and $\alpha_{3}=0.08$). (**a**) BER comparison between SISO-NOMA and OMA with η as 2. (**b**) BER comparison between SISO-NOMA and OMA with η as 4.

**Figure 5 sensors-24-07220-f005:**
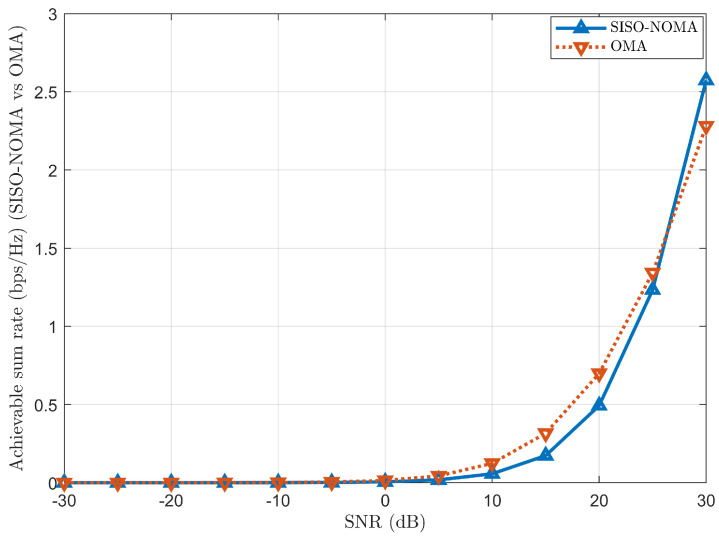
Achievable sum rate comparison between SISO-NOMA and OMA.

**Figure 6 sensors-24-07220-f006:**
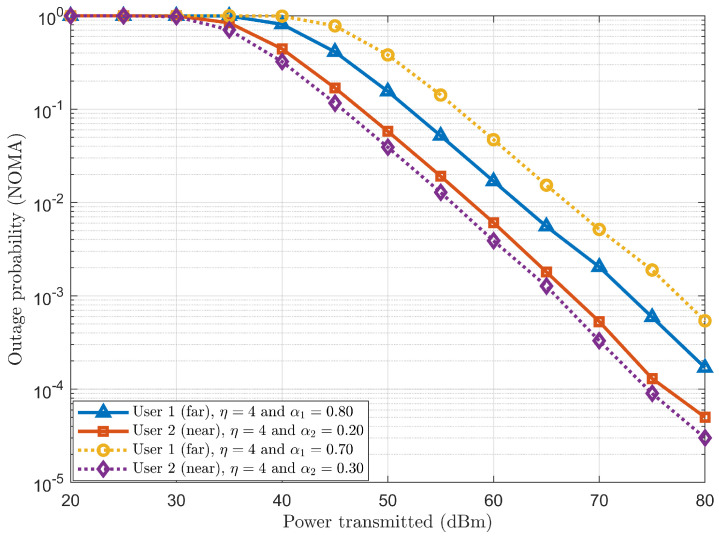
Outage probability of SISO-NOMA scheme.

**Figure 7 sensors-24-07220-f007:**
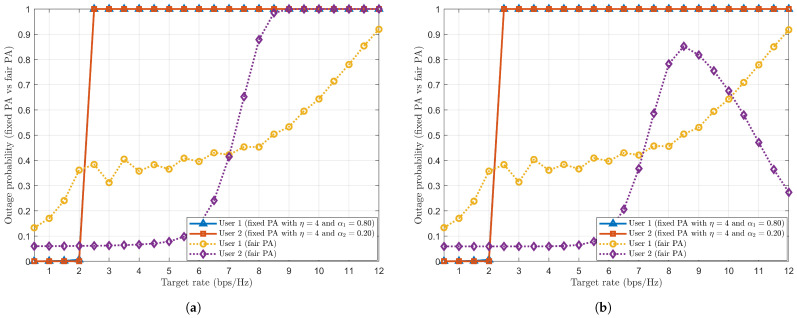
Outage probability of fair PA with a two-user scenario. (**a**) Outage probability of fair PA. (**b**) Improved outage probability of fair PA.

**Figure 8 sensors-24-07220-f008:**
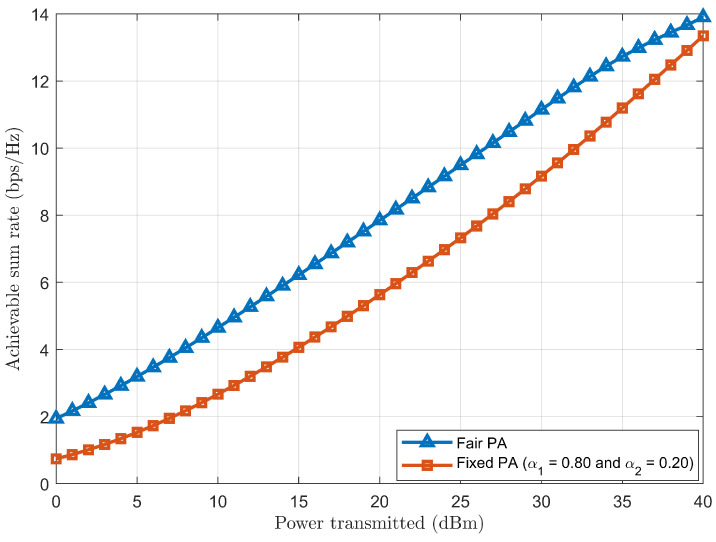
Achievable sum rate comparison between fair and fixed PAs.

**Figure 9 sensors-24-07220-f009:**
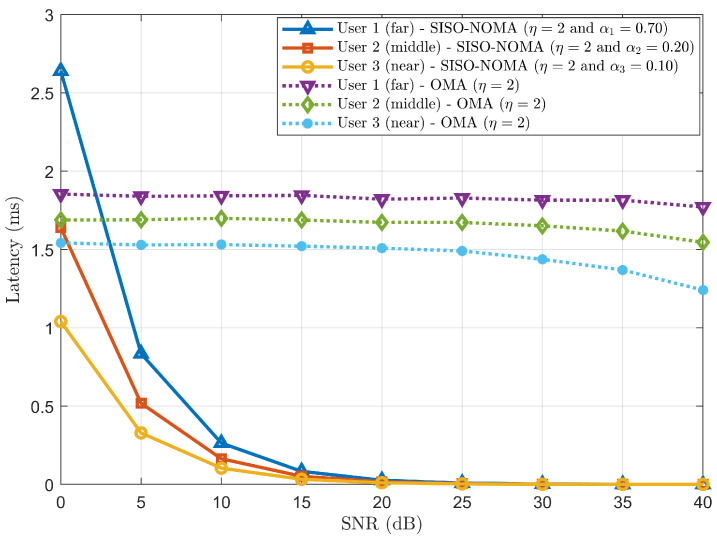
Latency comparison between SISO-NOMA and OMA with η = 2 and fixed PA coefficient ($\alpha_{1}=0.70$, $\alpha_{2}=0.20$ & $\alpha_{3}=0.10$).

**Figure 10 sensors-24-07220-f010:**
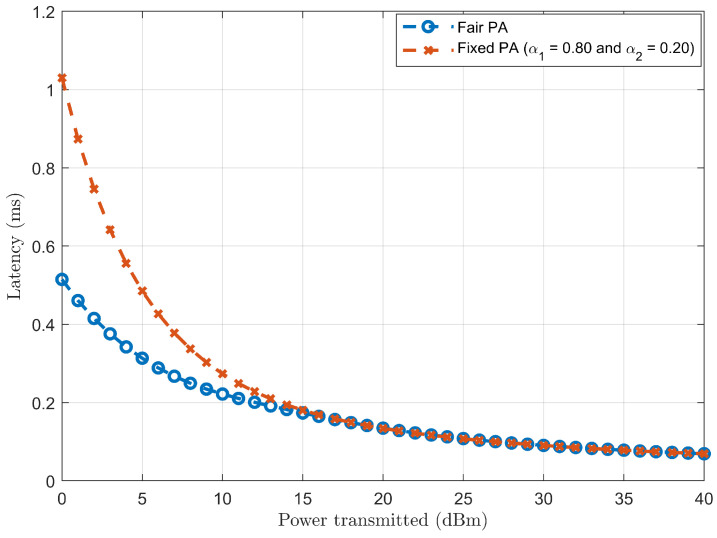
Latency comparison between fair and fixed PAs in SISO-NOMA.

**Figure 11 sensors-24-07220-f011:**
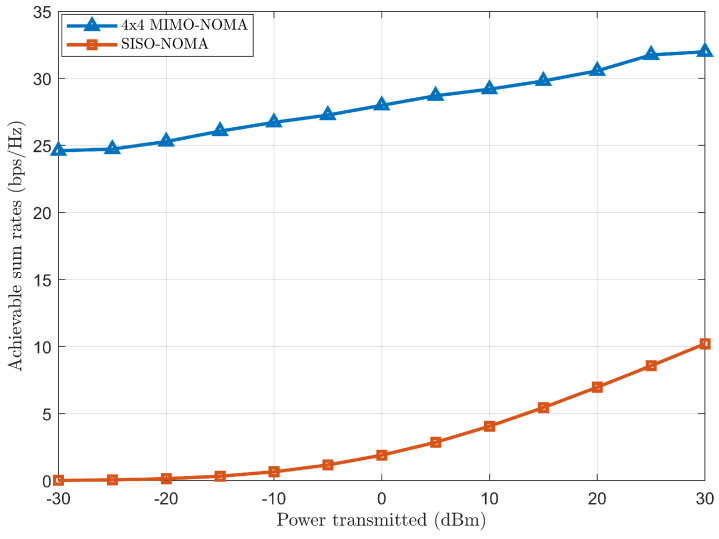
Achievable sum rate comparison between 4 × 4 MIMO-NOMA and SISO-NOMA.

**Figure 12 sensors-24-07220-f012:**
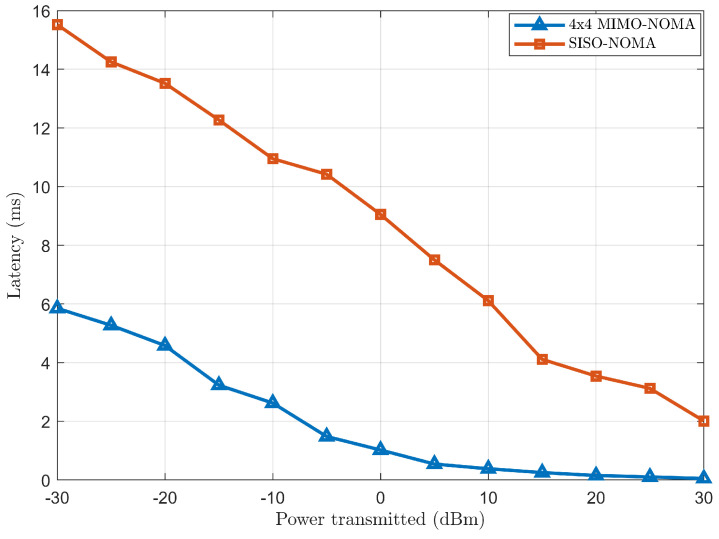
Latency comparison between 4 × 4 MIMO-NOMA and SISO-NOMA.

**Table 1 sensors-24-07220-t001:** Parameters used in the simulation.

Parameter	Value
Users 1, 2 and 3 distance from BS	1000, 500 and 200 metres, respectively
For a two-user scenario, PA coefficients ($\alpha_{1}$ & $\alpha_{2}$)	(70% & 30%) and (80% & 20%), respectively
For a three-user scenario, PA coefficients ($\alpha_{1}$, $\alpha_{2}$ & $\alpha_{3}$)	(70%, 20% & 10%) and (76%, 16% and 8%), respectively
Modulation scheme	Binary phase shift keying (BPSK)
Path loss exponent (η)	2 and 4
Channel	Rayleigh Fading
Number of OFDM subcarriers	128
Packet size	128 bytes
Noise	AWGN
System bandwidth	1 GHz
Power transmitted from BS	40 dBm

## Data Availability

No new data were created or analysed in this study. Data sharing is not applicable to this article.

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
