# Peer review of "A Study of Downlink Power-Domain Non-Orthogonal Multiple Access Performance in Tactile Internet Employing Sensors and Actuators"

_sensors, 2024, doi:10.3390/s24227220_

Round 1
Reviewer 1 Report
Comments and Suggestions for Authors
Please find the attached file.

Author Response
Please see attached pdf document.

Reviewer 2 Report
Comments and Suggestions for Authors
This is a very interesting paper on an important topic. The paper is well written and is technically sound. The authors provide an extensive set of results.
The background section and the references are ok. My only suggestion is to add these recent important papers written by G. Taricco on the topic:
G. Taricco Fair power allocation policies for power-domain non-orthogonal multiple access transmission with complete or limited successive interference cancellation
IEEE Access 2023
G. Taricco Optimum Power Allocation for HARQ Aided NOMA with Proportional Fairness on Fading Channels
IEEE Access 2024
Author Response
Please see attached pdf document.

Reviewer 3 Report
Comments and Suggestions for Authors
The novelty of this paper is questionable.
1. The authors propose a novel DL PD-NOMA scenario for Tacktile Internet (TI). However, there scenario itself cannot be a contribution. Moreover, in the TI restrictions in System model are not clear and the system looks like a common communication system. Therefore, the whole Section 3 is not novel and looks like first NOMA papers that are based on original Saito paper [1].
Estimations based on Shannon equation are straightforward. SINR, Sum Rate Analysis and Fair PA analysis have been already done for more complex systems with more complex signal constellations, see [2,3,4].
2. The authors claim that they consider TI but do not consider specific TI metrics such as latency.
3. It is strange to write that NOMA is a promising candidate for 5G if 5G New Radio is a working technology
1) Saito Y. et al. Non-orthogonal multiple access (NOMA) for cellular future radio access //2013 IEEE 77th vehicular technology conference (VTC Spring). – IEEE, 2013. – С. 1-5.
2) Zhang J. et al. Downlink non-orthogonal multiple access (NOMA) constellation rotation //2016 IEEE 84th Vehicular Technology Conference (VTC-Fall). – IEEE, 2016. – С. 1-5.
3) Khorov E. et al. Prototyping and experimental study of non-orthogonal multiple access in Wi-Fi networks //IEEE Network. – 2020. – Т. 34. – â„–. 4. – С. 210-217.
4) Sangdeh P. K. et al. A practical downlink NOMA scheme for wireless LANs //IEEE transactions on communications. – 2020. – Т. 68. – â„–. 4. – С. 2236-2250.
Comments on the Quality of English Language
Grammar mistakes should be fixed
Author Response
Please see attached pdf document.

Round 2
Reviewer 3 Report
Comments and Suggestions for Authors
It is great that the authors added latency analysis.
However, in my opinion there is still not clear why it is so necessary to mention the term Tactile Internet in this paper. Tactile Internet has quite strict requirements for transmission latency and BER which are not considered in this paper.
Moreover, the authors consider too simple systems, where there are no more than 3 users. In real 5G systems there are more users and the main bottleneck for Tacktile Internet data is queueing delay. So, it is necessary to model scheduling processes in your simulation.
Also, it is difficult to find SISO 5G systems, most of them are MIMO, so, it is better for the authors to compare their results not only with OMA systems, that have been done dozens of times in a previous decade, but also with MIMO.
Comments on the Quality of English LanguageEnglish is fine, minor edditing is required
